

# Susceptibility of Human Oral Squamous Cell Carcinoma (OSCC) H103 and H376 cell lines to Retroviral OSKM mediated reprogramming

Nalini Devi Verusingam[1], Swee Keong Yeap[2,8], Huynh Ky[3], Ian C. Paterson[4], Suan Phaik Khoo[5], Soon Keng Cheong[1,6], Alan H.K. Ong[1] and Tunku Kamarul[7]

[1] Faculty of Medicine and Health Sciences, Universiti Tunku Abdul Rahman, Selangor, Malaysia
[2] Institute of Bioscience, Universiti Putra Malaysia, Selangor, Malaysia
[3] College of Agriculture and Applied Science, Cantho University, Vietnam
[4] Department of Oral Biology & Biomedical Sciences, Faculty of Dentistry, Universiti Malaya, Kuala Lumpur, Malaysia
[5] School of Dentistry, International Medical University, Kuala Lumpur, Malaysia
[6] Majlis Kanser Nasional (MAKNA) Cancer Research Institute, Kuala Lumpur, Malaysia
[7] Tissue Engineering Group, National Orthopaedic Centre of Excellence for Research and Learning, Department of Orthopaedic Surgery, Faculty of Medicine, University Malaya, Kuala Lumpur, Malaysia
[8] Current affiliation: China-ASEAN College of Marine Sciences, Xiamen University Malaysia, Selangor, Malaysia

Corresponding author
Alan H.K. Ong, onghk@utar.edu.my

## ABSTRACT

Although numbers of cancer cell lines have been shown to be successfully reprogrammed into induced pluripotent stem cells (iPSCs), reprogramming Oral Squamous Cell Carcinoma (OSCC) to pluripotency in relation to its cancer cell type and the expression pattern of pluripotent genes under later passage remain unexplored. In our study, we reprogrammed and characterised H103 and H376 oral squamous carcinoma cells using retroviral OSKM mediated method. Reprogrammed cells were characterized for their embryonic stem cells (ESCs) like morphology, pluripotent gene expression via quantitative real-time polymerase chain reaction (RT-qPCR), immunofluorescence staining, embryoid bodies (EB) formation and directed differentiation capacity. Reprogrammed H103 (Rep-H103) exhibited similar ESCs morphologies with flatten cells and clear borders on feeder layer. Reprogrammed H376 (Rep-H376) did not show ESCs morphologies but grow with a disorganized morphology. Critical pluripotency genes Oct4, Sox2 and Nanog were expressed higher in Rep-H103 against the parental counterpart from passage 5 to passage 10. As for Rep-H376, Nanog expression against its parental counterpart showed a significant decrease at passage 5 and although increased in passage 10, the level of expression was similar to the parental cells. Rep-H103 exhibited pluripotent signals (Oct4, Sox2, Nanog and Tra-1-60) and could form EB with the presence of three germ layers markers. Rep-H103 displayed differentiation capacity into adipocytes and osteocytes. The OSCC cell line H103 which was able to be reprogrammed into an iPSC like state showed high expression of Oct4, Sox2 and Nanog at late passage and may provide a potential iPSC model to study multi-stage oncogenesis in OSCC.

## INTRODUCTION

Oral cancer is the sixth most common malignancy worldwide and is more prevalent in the developing than developed countries (*Warnakulasuriya, 2009*). More than 90% of oral cancer cases are classified as Oral Squamous Cell Carcinoma (OSCC) which is a malignant epithelial cancer that arises from oral keratinocytes (*Scully & Bagan, 2009*). As there are no reliable diagnostic markers and early stage OSCC is asymptomatic, patients with OSCC are often presented for treatment with advanced stage cancer resulting in a poor prognosis (*Warnakulasuriya, 2009*; *Markopoulos, Michailidou & Tzimagiorgis, 2010*).

In spite of the advancement in molecular based detection of cancer stages (*Bose et al., 2012*; *Rhandawa & Archaraya, 2015*), the underlying biological mechanisms that take place within the OSCC progression have not been well established. Existing models for OSCC study derived from xenograft of primary tumours have been problematic as *in-vitro* studies having resulted in a low percentage of cell number and induced mutations with prolonged culture. Furthermore, human cell models of OSCC from tumour cell lines generally can only mimic the advanced tumour state which does not allow the different stages of cancer to be monitored and studied (*Shirako et al., 2015*). Hence, these unresolved issues necessitate an approach to establish human cancer models that recapitulate OSCC progression.

Originally, induced pluripotent stem cells (iPSCs) were established from adult human somatic stem cells through the transient expression of Yamanaka's Oct4, Sox2, KLf4 and c-Myc (OSKM) transcription factors reprogramming technology (*Takahashi et al., 2007*) and used as a promising stem cell source to overcome ethical and immune rejection issues that often surfaced when using human Embryonic Stem Cells (hESCs) (*Yamanaka, 2007*). Subsequent studies demonstrated that reprogrammed cancer cells were able to exhibit pluripotent capacity and differentiation tendency, which were distinct from that of their parental cells with much of the observable changes owing to epigenetic effects of reprogramming (*Miyoshi et al., 2009*; *Carette et al., 2010*; *Mahalingam et al., 2012*; *Gandre-Babbe et al., 2013*; *Kim et al., 2013*; *Zhang et al., 2013*; *Choong et al., 2014*; *Koga et al., 2014*; *Kotini et al., 2015*; *Lee et al., 2015*). Interestingly, while the majority of the reprogrammed cancer cells showed either lower or a loss of tumorigenicity, *Carette et al., 2010* and *Gandre-Babbe et al., 2013* indicated a re-establishment of the oncogenic dominance over the pluripotent cell phenotype when the reprogrammed cell types were differentiated into hematopoietic lineage cells and Kim and colleagues (*2013*) showed that the reprogrammed human pancreatic cancer cells were able to recapitulate its cancer progression from early to late stage of cancer development upon differentiation. In contrast, MCF-7 breast cancer cell line which was reprogrammed via retroviral-OSKM method, failed to show distinct pluripotent signals and was unable to differentiate into 3 primary germ layers but displayed instead a typical cancer stem cell (CSC) phenotype (*Corominas-Faja et al., 2013*). These reprogrammed phenotypes of cancer cells have made the study of cancer development and underlying cellular systems more feasible, much of which were not previously encountered from any available cancer models especially in addressing cancer progression, discovery of cancer specific biomarkers and more effective therapy (*Lang, Shi & Chin, 2013*; *Lee et al., 2015*; *Kim & Zaret, 2015*).

To the best of our knowledge, no studies have previously reported on the reprogramming of OSCC cancer cells and therefore in the present study, we examined the susceptibility of two OSCC cell lines to be reprogrammed into pluripotency and their respective pluripotent gene expression patterns.

## MATERIALS AND METHODS

### Cell culture

Human Oral Squamous Cell Carcinoma cell lines (OSCC), H103 (STNMP Stage I) and H376 (STNMP StageIII), were obtained from Prof Ian Patterson, University Malaya, Kuala Lumpur, Malaysia. Cell lines were cultured in DMEM/F12 supplemented with 10% fetal bovine serum (FBS) (Gibco/Invitrogen, Grand Island, NY, USA) and Hydrocortisone (0.5 ug/ml) (Sigma-Aldrich, St. Louis, MO, USA) at 37 °C in the presence of 5% $CO2$. Reprogrammed OSCC cells were maintained in conditioned hESC medium consisting of DMEM/F12 (Gibco/Invitrogen, Grand Island, NY, USA), supplemented with 20% knock-out serum replacement (Gibco/Invitrogen, Grand Island, NY, USA), 0.1 mM nonessential amino acids (Gibco/Invitrogen, Grand Island, NY, USA), 4 mM L-glutamine (Sigma-Aldrich, St. Louis, MO, USA), 10 ng/mL fibroblast growth factor (bFGF) (Invitrogen, Carlsbad, CA, USA) and 0.1 mM 2-mecaptoethanol (Sigma-Aldrich, St. Louis, MO, USA). The medium was changed every 48 h.

### Retrovirus production and infection

Retroviral vectors were produced via packaging cell lines, 293FT cell lines (human embryonal kidney cells) (Thermo Fisher Scientific, Waltham, MA, USA) using the Yamanaka Factors (Oct4, Sox2. Klf4, c-Myc). Vectors pMX-based retroviral h*Oct4* (Plasmid 17217) (Addgene, Cambridge, MA, USA), h*Sox2* (Plasmid 17218) (Addgene, Cambridge, MA, USA), h*Klf4* (Plasmid 17219) (Addgene, Cambridge, MA, USA), h*c-Myc* (Plasmid 17220) (Addgene, Cambridge, MA, USA), retroviral gag–pol packaging plasmid (Plasmid 8449) (Addgene, Cambridge, MA, USA), VSV-G expression plasmid (Plasmid 8454) (Addgene, Cambridge, MA, USA) and pMX-GFP (Cell Biolabs, San Diego, CA, USA) used in these experiments were provided by Dr. Shigeki Sugii, DUKE-NUS Graduate Medical School, Singapore. Human embryonal kidney cell lines were plated at $3.6 \times 10^6$ (70%–80% confluency) one day before transduction. The packaging cell lines were then transfected with Retro-GFP/OSKM vectors along with the transfection reagent LIPOFECTAMINE 2000 (Life Technologies, USA) according to the manufacturer's protocol. The supernatant was collected at 48 h post-transfection and filtered with PVDF (45 uM) (Merck Milipore, Billerica, MA, USA). Polybrene (0.5 ul/ml from 10 mg/ml) was then added into the supernatant containing Retroviral Vectors to enhance the transduction efficiency within the target cell types. Presence of selection marker, green fluorescent protein (GFP) in H103 and H376 were counted manually using the cell count option in ImageJ program.

### Retroviral infection and iPSCs cell generation

Human Oral Squamous Carcinoma Cells (H103 and H376) were seeded at $7.5 \times 10^4$ cells/well on 6 well plates, 24 h before transduction. Equal amounts of supernatant

containing each of the four retroviruses were collected and mixed prior to transduction of OSCC cell lines. Infected cells were incubated overnight. The medium was changed at 24 h post-infection to H103 and H376 specific medium. Transduced cancer cells were then subjected to spinfection at 1 h and 30 min (32 °C) before placing the plates into a hypoxic incubator (5% $O_2$). At day 3 post infection, OSCC cell lines were harvested by trypsinization and re-plated with $1 \times 10^4$ cells/well on 6 well plates containing irradiated Mouse Embryonic Feeder Layer (MEF) (GlobalStem, Rockville, MD, USA). After 24 h, the medium was replaced with hESCs medium. Fifteen days after transduction, formed colonies were picked up and transferred onto a new mouse embryonic fibroblast (MEF) feeder layer (GlobalStem, Rockville, MD, USA). These colonies were propagated at least up to five passages before being subjected to pluripotency characterization.

## Gene expression assessment in OSCC-induced pluripotent stem cells (iPSCs)

Ribonucleic acid (RNA) was extracted from parental H103 and H376 and respective reprogrammed counterparts using RNeasy Mini Kit (Qiagen, Hilden, Germany) according to the manufacturer's protocols. Total RNA was then converted into complementary deoxyribonucleic acid (cDNA) via reverse transcription (QuantiTech Reverse Transcriptase Kit).The fluorescent signals from each sample were plotted against cycle numbers which represented the accumulation of product over the duration of the quantitative real-time PCR (qRT-PCR) experiment. Quantitative real-time PCR amplification efficiency generated with primers (Table S1) were optimized using hESC, a commonly used gold standard for pluripotency characterization. Beta-actin (ACTB) was used as a house keeping gene in this experiment. Briefly, the condition used in this experiment comprises initial denaturation at 95 °C for 15 min, followed by 40 cycles of denaturation at 94 °C for 30 s with annealing at 60 °C for 30 s and extension at 72 °C. Following qRT-PCR, fluorescence data collection was performed during extension. Real-time PCR was performed with iQ5 Bio-Rad qPCR machine (Bio-Rad, Hercules, CA, USA) using Quantitect Sybr Green PCR Master Mix (Qiagen, Hilden, Germany). iQ5 Optical System Software, Version 2.0 was used for the analysis (Bio-Rad Laboratories, Hercules, CA, USA).

## Immunofluorescense staining

Cells were fixed using 4% v/v paraformaldehyde (Sigma-Aldrich, St. Louis, MO, USA), washed three times with PBS containing 1% BSA and permeablized using Perm Buffer (BD Biosciences, San Jose, CA, USA) for 15 min at room temperature (Intracellular markers). After permeabilization, cells were blocked with PBS containing 1% BSA for 1 h at room temperature after the blocking solution was removed and cells were washed three times with PBS. Cells were then incubated with conjugated antibodies in PBS containing 1% BSA overnight at 4 °C. Antibodies used were at 1:50 dilution factor for Oct4-PE (BD Biosciences, San Jose, CA, USA), Sox2-PE (BD Biosciences, San Jose, CA, USA) Nanog-ALexa Fluor 488 (BD Biosciences, San Jose, CA, USA), and Tra-1-60-PE (BD Biosciences, San Jose, CA, USA). After the overnight incubation, cells were washed three times with PBS and stained with DAPI ANTIFADE GOLD (Invitrogen, Carlsbad, CA, USA) prior to viewing under Zeiss Imager A.1 Fluorescence Microscope (Carl Zeiss, Oberkochen, Germany).
## Embryoid bodies (EB's) formation

Rep-H103 cells were seeded onto ultra-low attachment plates (Corning) containing commercialize embryoid bodies medium (Millipore, Billerica, MA, USA). Transferred cells were grown in suspension for 8 days. The medium was consistently changed every 2–3 days up to 8 days without disrupting the EBs. EBs formed were carefully collected for immunofluorescence staining to determine the presence of three germ layers specific markers (Ectoderm: OTX2, Sox1; Endoderm: Sox17, Gata4; Mesoderm: Brachyury) according to the manufacturer's instructions (Human Three Germ Layer 3-Color Immunostainining Kit) (R&D Systems, Inc., Minneapolis, MN, USA). Stained cells were observed under Zeiss Imager A.1 Fluorescence Microscope (Carl Zeiss, Oberkochen, Germany).

## Directed differentiation of human H103 iPSCs like cells

Since H103 cell line is of the ectoderm and endoderm lineage (*Jones & Klein, 2013*), reprogrammed H103 was subjected to directed differentiation into adipocytes and osteocytes which is of the mesoderm lineages in order to access its differentiation potential.

## Osteogenic assay

Osteogenic differentiation medium (ODM) (*Erenpreisa & Cragg, 2013*) was used to induce mineralization or osteogenesis in reprogrammed H103. IPSC-like cells were plated on gelatine coated six-well plates and cultured in 3ml of ODM consisting of 90% DMEM/F12 (Gibco/Invitrogen, Grand Island, NY, USA), 10% FBS (Gibco/Invitrogen, Grand Island, NY, USA) supplemented with 10nM dexamethasone (Sigma-Aldrich, St. Louis, MO, USA), 20 mM β-glycerol phosphate (Sigma-Aldrich, St. Louis, MO, USA), and 50 μM L-ascorbic acid (Sigma-Aldrich, St. Louis, MO, USA). The medium was changed every 2–3 days. After 21 days of incubation at 37 °C, 5% $CO_2$, cells were stained in Alizarin Red for visualization of calcium deposits. Stained cells were then evaluated under Eclipse TS100 inverted microscope (Nikon, Japan) and images were captured for analysis. Presence of mineralized osteoblasts indicated bright orange-red precipitate.

## Adipogenic assay

Colonies plated on gelatine coated six-well plates and were cultured in 3ml of adipogenic differentiation medium (ADM) (*Erenpreisa & Cragg, 2013*) per well. ADM consisted of 90% DMEM/F12 (Gibco/Invitrogen, Grand Island, NY, USA), 10% FBS (Gibco/Invitrogen, Grand Island, NY, USA) supplemented with 0.5 μM dexamethasone, 0.5 μM isobuthylmethylxanthine and 50 μM indometacin. The medium was changed every 2–3 days. After 21 days of incubation at 37 °C, 5% $CO_2$, cells were stained in Oil Red O for visualization of lipid droplets. Stained cells were then viewed under Eclipse TS100 inverted microscope (Nikon, Japan) and images were captured for analysis. Mature adipocytes containing intracellular lipid vesicles were stained bright red.

## Statistical analysis

Statistical data analysis was carried out with Paired $t$-Tests to compare the quantitative outcome of parental, reprogrammed counterparts at passage 5 and reprogrammed

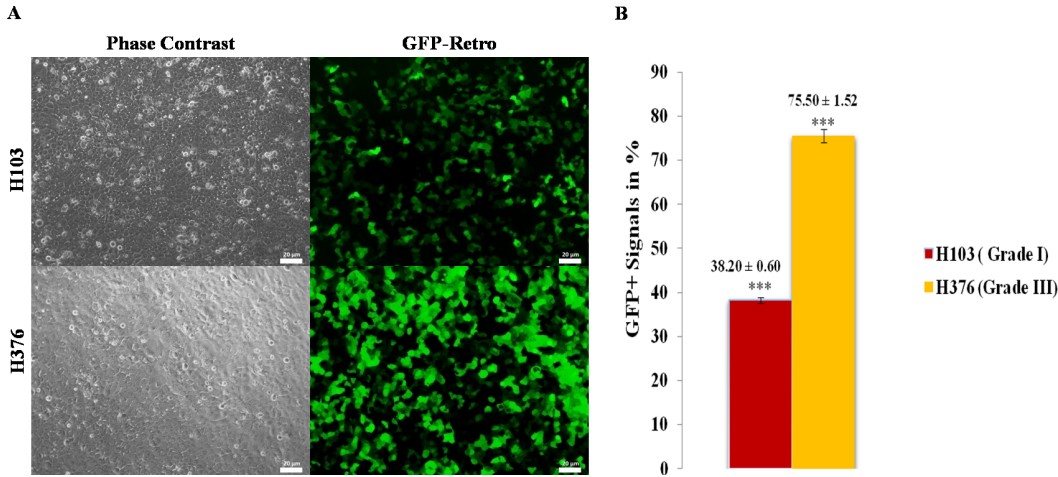

**Figure 1** **Transduction efficiency of retroviruses in OSCC.** (A) Oral squamous carcinoma cell lines (H103 and H376) were introduced with ecotropic pMXs retroviruses containing GFP cDNA. The upper panel shows the images of bright-field and fluorescent microscope, Zeiss Axiovert inverted microscope, original magnification: 20×. (B) The lower panel shows the percentages of cells transduced with GFP. Data are expressed as mean ± standard deviation (SD).

counterparts at passage 10 of H103 and H376 cell lines using SPSS, Software version 22.0 (IBM Corp, USA). All tests were conducted at 95% confidence level and all data were presented as mean ± standard error of mean SEM. The differences were considered significant at $P < 0.05$.

## RESULTS

### Transfection efficiencies in oral squamous cells carcinoma cell lines

Green Fluorescence Protein (GFP) serves as an internal control and used for evaluating transduction efficiency. Uptake of vector pMX-GFP was tested using the highly transfectable human embryonic kidney cells (293FT) (Fig. S1). Subsequently, transduction efficiencies were examined in OSCC cell lines using vector pMX-GFP (16.5 μg) which encodes for green fluorescent protein signals to confirm the uptake of transgenes prior to OSKM transduction. GFP signals were detected in H103 and H376 at 48 h confirming the uptake of transgenes (Fig. 1A). The transduction efficiency of 75.50% ± 1.52 and 38.20% ± 0.60 obtained from H376 and H103 cells respectively showed an almost 2-fold increase in H376 (Fig. 1B). Negative control was performed to access the potential influence of reprogramming on the viability of the cells (Fig. S2).

### Reprogramming of oral squamous cell carcinoma cell lines

Clones were picked approximately two weeks after transduction (Fig. 2A) in both H103 and H376 cells. Distinct morphological patterns and changes were observed between the parental cancer cells and their reprogrammed counterparts (Fig. 2B i–iii and 2C i–iv). Only Rep-H103 clones were able to be passaged above passage 5 (Fig. 2B-iv) while clones derived from Rep-H376 differentiated at passage 2 onwards and were unable to sustain iPS-like cell morphology (Fig. 2C-v). Although higher GFP transfection capacity was achieved in H376,

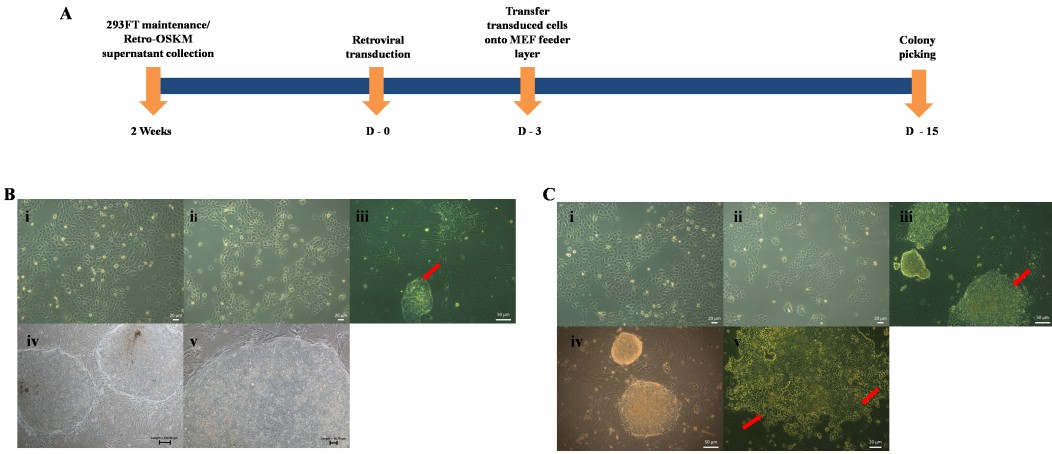

**Figure 2** **Induction of Pluripotent cells from OSCC.** (A) Timeline of iPSCs induction. (B-i) Morphology of H103. (B-ii) Morphology of transduced H103 at 24 h post infection. (B-iii) Emergence of iPSCs like colonies from reprogrammed H103 on MEF feeder layer at day 15 (Red arrow). (B-iv-v) Images of established stable H103 iPSCs like cells at passage number 10. (C-i) Morphology of H376. (C-ii) Morphology of transduced H376 at 24 h post infection. (C-iii) Emergence of iPSC like colonies from H376 on MEF feeder layer at day 15 (Red arrow). (C-vi) Image of iPSC like cells from H376 at P2. (C-v) Differentiated reprogrammed H376 (Red arrows), collected at P5 and P10 for real time expressions analysis.

stable clones were successfully generated only from H103 cell line as these clones were able to be expanded up to passage 20 and still maintained an ESC-like morphology (Fig. 2B iv–v). Morphologies of all derived clones from H103 are highly distinct from its parental cells with clear borders on the mouse embryonic fibroblast feeder layer, high nucleus to cytoplasm ratio and the colonies displayed small cells morphology with spaces between them (Fig. 2B iv–v).

## Differential gene expression

Overall, the expressions of endogenous Oct4, Sox2 in Rep-H103 cells were much higher than that of Rep-H376. However, suppression of Klf4 and c-Myc was seen in both reprogrammed cells (Fig. 3). Pluripotent marker expression levels of the reprogrammed cells relative to the parental counterparts which were quantified using qRT-PCR (Fig. 3A) showed down-regulation of Oct4 expression in rep-H103 at passage 5 but increased with a 2.80 fold change at passage 10. Expression of Sox2 (Fig. 3B) showed gradual up-regulation in Rep-H103 at passage 5 and passage 10 with 55 and77 fold change respectively. Oncogenic Klf4 (Fig. 3C) gene expression in reprogrammed H103 was significantly down regulated upon reprogramming and the level of expression maintained throughout passage 5 and passage 10. Similar gene expression pattern as Klf4 was observed in c-Myc gene (Fig. 3D) but at a much lower expression level. Nanog expression was also up-regulated gradually across passage 5 with 2.84 fold change to passage 10 with 7.07 fold change relative to its parental counterpart (Fig. 3E).

In the case of Rep-H376, Oct4 expression showed an initial reduction at passage 5 of −4.35 fold change but was up-regulated at passage 10 with 1.07 fold change (Fig. 3A). Gradual increase of Sox2expression (Fig. 3B) from passage 5 to passage 10 was observed in

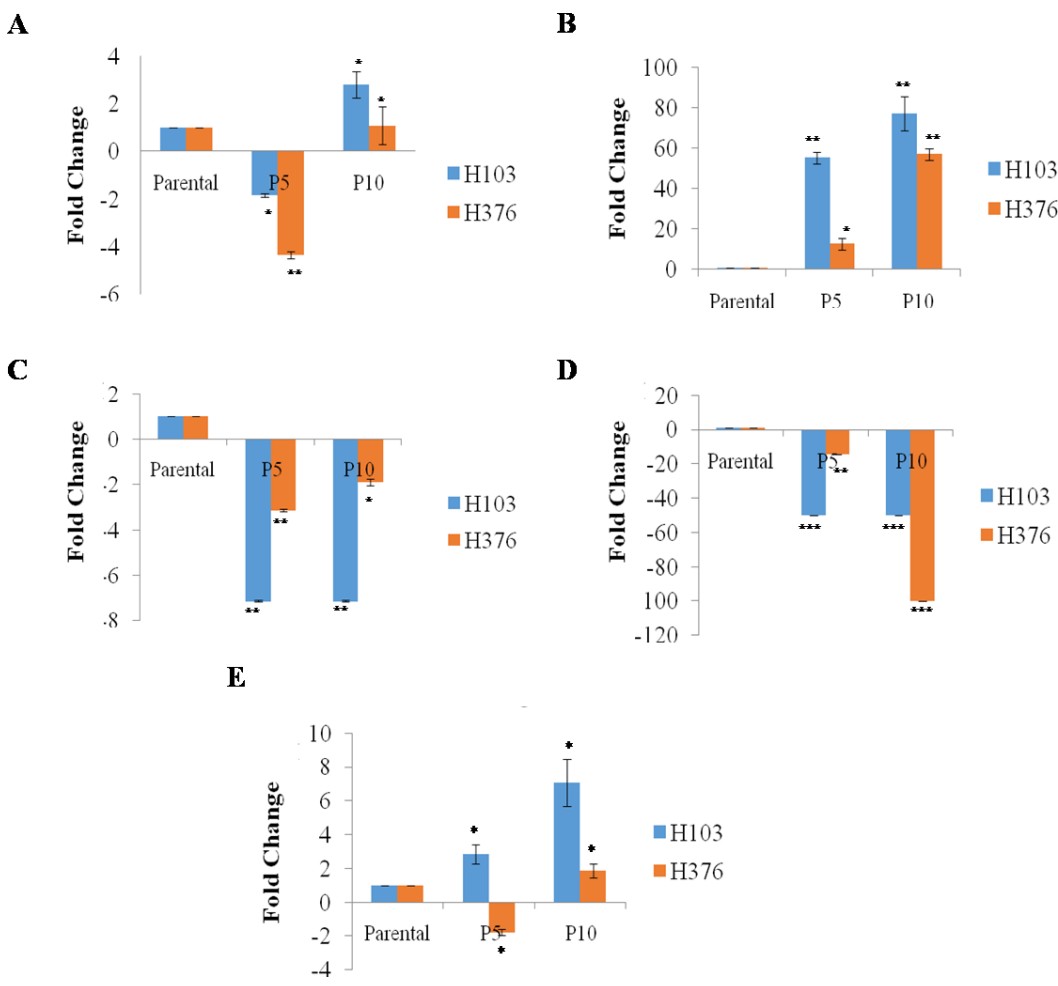

**Figure 3** *In-Vitro* **mRNA expression of pluripotent genes in reprogrammed H103 and H376 relative to parental (n-3).** (A–E) Data presented as mean ± SEM. Statistical differences are indicated with * for $P < 0.05$, ** for $P < 0.01$ and *** for $P < 0.001$ using paired $t$-test.

Rep-H376, a similar pattern to that of Rep-H103. Down-regulation of Klf4 expression was observed at passage 5 but slightly increased from −3.13 fold change to −1.89 respectively at passage 10 (Fig. 3C). Expression of c-Myc was almost not detected with −14.29 fold change at passage 5 and −100 at passage 10 (Fig. 3D). Nanog expression was found reduced in Rep-H376 at passage 5 with −1.79 fold change and showed little increase at passage 10 with 1.86 fold change (Fig. 3E).

## Expression of pluripotency associated transcription factors

Rep-H103 continuously proliferated as adherent, flat colonies under feeder conditions and captured satisfactory pluripotent signals from real-time PCR analysis compared to reprogrammed H376. Therefore, successful Rep-H103 cells were characterised further for its pluripotency protein expression via immunofluorescence analysis. Rep-H103 expressed the pluripotency markers of Oct4, Sox2, Nanog and Tra-1-60 indicating a distinct difference between H103 derived iPS-like cells (Fig. 4A) over its parental cancer cell line (Fig. 4B).

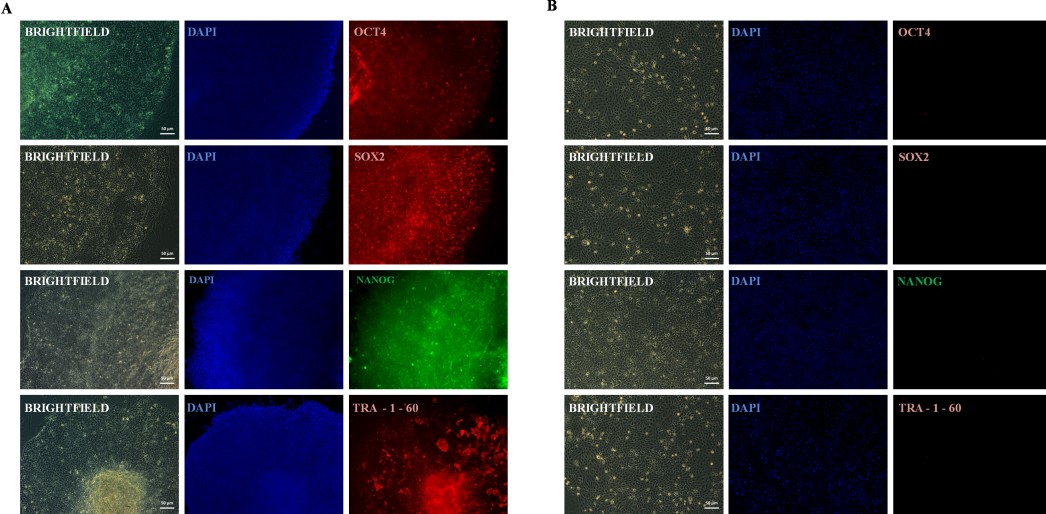

**Figure 4   Expression of pluripotent marker.** (A) REP-H103 expressed markers common to pluripotent cells including OCT4, SOX2, NANOG and TRA-1-60. (B) Pluripotency markers (OCT4, SOX2, NANOG and TRA-1-60) were not expressed in the parental cancer cell.

## Differentiation: embryoid body formation

Three dimensional (3D) sphere-shaped structures in suspension culture formed within a week in EB specific medium. At day 8, EBs were then evaluated for the presence of three germ layers via immunofluorescence staining. EBs was also induced from the parental cell line of H103 to serve as control (Fig. 5A-i). Rep-H103 showed morphologically compact round borders EB (Fig. 5A-ii). As H103 cell line was derived from both ectodermal and endodermal lineage expressions, signals from these two lineages were expected to be detected in EBs from both the parental and the reprogrammed counterpart (Figs. 5B-i and 5B-ii). However, mesoderm (Fig. 5B-iii) expression was only detected in EBs derived from Rep-H103 indicating the cross lineage differentiation ability of the Rep-H103 upon reprogramming into an iPS-like cell.

## Directed differentiation assay

Crystal formation was observed during the osteogenesis process whereby the volume of the matrix mineralization increased during the 21 days of incubation as indicated from the Alizarin Red S stained calcium deposits (Fig. 6A-ii). In addition, tiny vesicles containing lipid droplets had formed in the cytoplasm of the cells after 21 days of incubation. The accumulation of lipid droplets was stained positive with Oil Red O staining (Fig. 6A-iv).

## DISCUSSION

Overall, different responses towards retroviral-OSKM mediated reprogramming were observed in two different grade OSCC cell lines (H103-STNMP Stage I and H376-STNMP Stage III). Rep-H103 exhibit ESCs like morphology, are distinguishable from the parental cell line and generated H103 iPS-like cells could undergo *in-vitro* expansion more than 20 passages. The ESC-like morphological changes in Rep-H103 corresponded to other

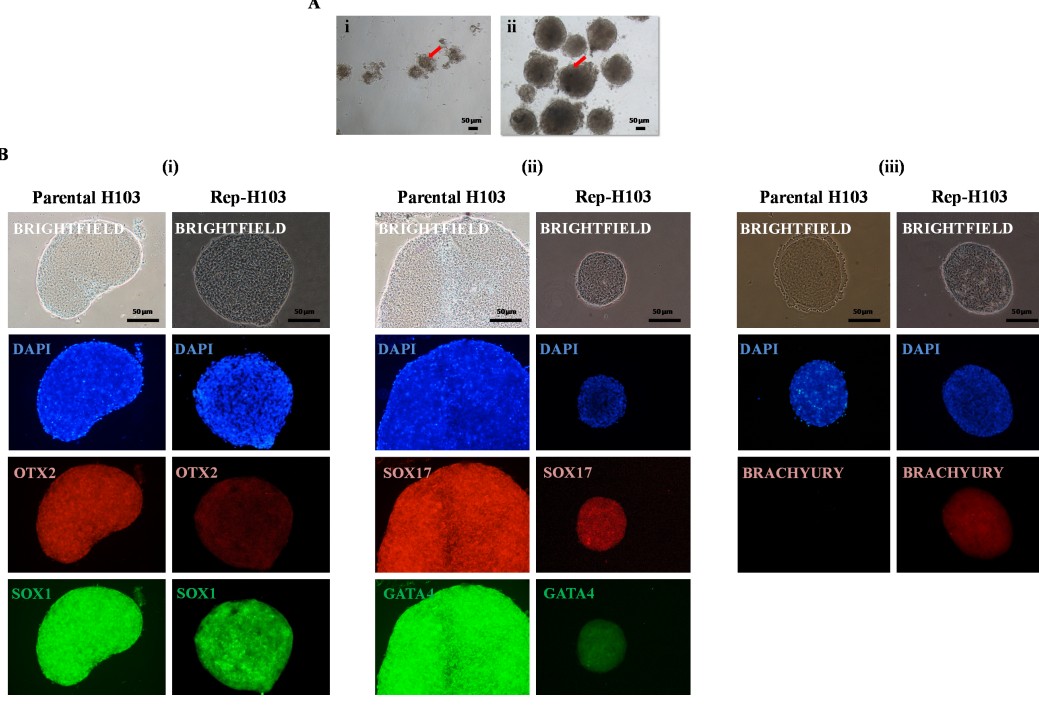

**Figure 5  Embryoid bodies (EBs) formation and immunofluorescence staining.** Representative images of (A-i) parental H103-EB cells and (A-ii) reprogrammed H103-EB. The better showed compact structure and round borders-EB exhibited by reprogrammed H103-EB which is lacking in parental H103-EB structure (Red arrow). Nikon inverted microscope, original magnification: 10×. Immunostaining analysis indicates the presence of three germ layers of (B-i) ectoderm—OTX2/SOX1, (B-ii) endoderm—GATA4/SOX17 and (B-iii) mesoderm—BRACHYURY.

reprogrammed cancer cell lines including chronic myeloid leukemia (CML) cancer cell line (KBM7 cells) (*Carette et al., 2010*), lung cancer cells (*Mahalingam et al., 2012*), liver cancer cells (*Zhang et al., 2014*) melanoma cells (*Lin et al., 2008*) and osteosarcoma (*Zhang et al., 2013*; *Choong et al., 2014*). On the other hand, reprogrammed H376 tended to differentiate into its original phenotype and the ESC-like features were no longer observable after passage 2 (Fig. 2C-v).

Although Rep-H103 cell had successfully acquired pluripotent ectopic expressions and sustained its pluripotency potentiality, it possessed lower GFP transgene uptake efficiency as compared to that of Rep-H376 (Fig. 1).This feature is consistent with a previous reported study in which, reprogramming of four different osteosarcoma cell lines (Saos-2, MG-63, U-2 OS and G-292) via Retroviral-OSKM mediated system demonstrated that U-2 OS cell line, which was highly responsive to GFP transduction, eventually lost its pluripotency capacity upon reprogramming and could not be maintained in prolonged *in-vitro* culture. Presence of various intrinsic factors between the osteosarcoma cancer cell lines was hypothesized to contribute to multiple responses towards reprogramming (*Choong et al., 2014*). As such, depending on the cell type, uptake of OSKM transgenes may not necessarily be correlated with transduction efficiency.

**A**

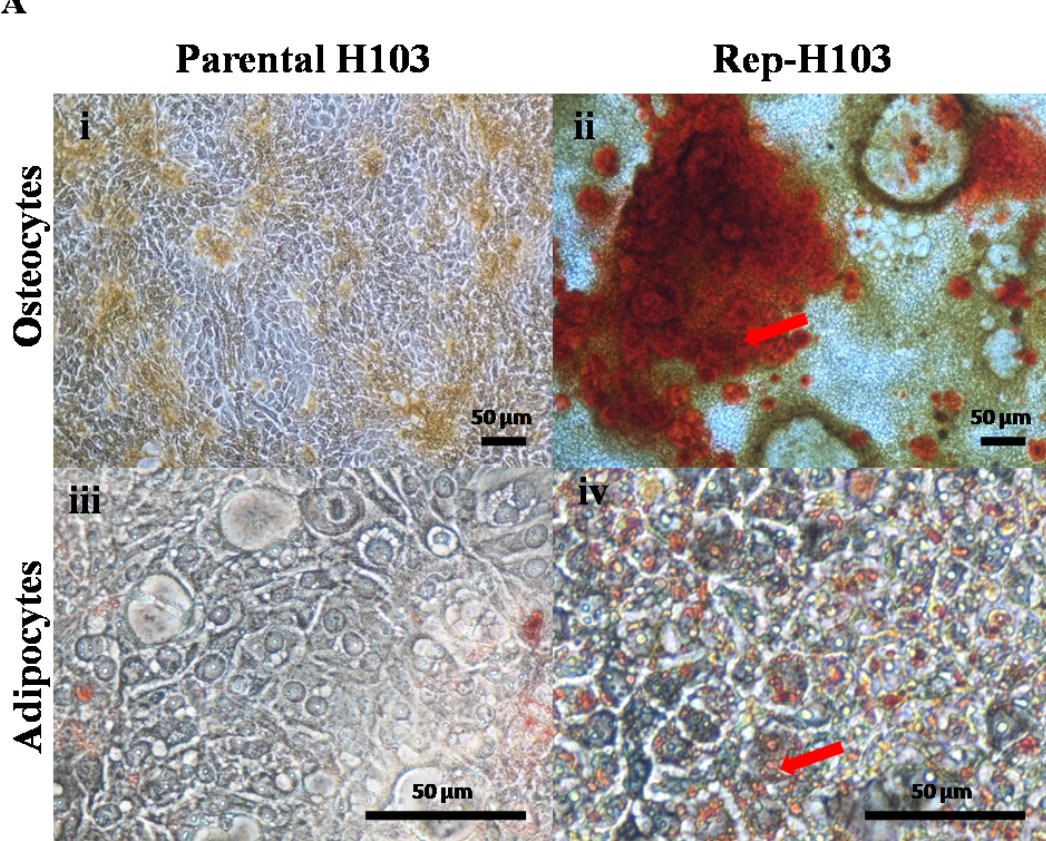

**Figure 6** **Directed differentiations into osteocytes and adipocytes.** (A-i) Control parental H103 cells showing negative for the Alizarin red staining (A-ii) H103 iPSCs like cells showing positive for the Alizarin red staining indicating calcium deposits (Red arrow). (A-iii) Negative staining for Oil-Red-O in control parental H103 cells. (A-iv) Adipogenesis induced lipid droplets observed red in colour stain after Oil-O-Red staining. (Red arrow) indicates the tiny lipid droplets. Nikon inverted microscope, original magnification: (A-i & ii) 10×, (A-iii & iv) 40×.

Reprogramming resistance was observed in H376 cell line in which the expressions of key pluripotent genes were insignificant and the ESC-like morphological features eventually diminishes with subsequent passage. Reprogramming roadblocks contributed by diverse molecular properties and biophysical nature of the cell type could result in inefficient reprogramming (*Vierbuchen & Wernig, 2012*). Notably, the tumour suppressor gene, p53 which safeguards the cellular genome integrity was shown to lower reprogramming efficiency and kinetics by removing DNA damaged cells at the early stages of the reprogramming stepwise process via apoptosis (*Vierbuchen & Wernig, 2012*; *Spike & Wahl, 2011*). Although both H103 and H376 harbours the mutant p53 gene, H376 carries a p53 gene, a nonsense mutation that expresses the truncated form of the protein which basically does not show any detectable mutant p53 expression (*Yeudall et al., 1995*). Despite the fact that the mutated form of p53 gene may provide a more favourable condition for reprogramming cancer cells (*Ebrahimi, 2015*; *Vierbuchen & Wernig, 2012*; *Spike & Wahl, 2011*), the presence rather than the absence of the mutant p53 expression has been

reported to enhance reprogramming efficiency (*Sarig et al., 2010*; *Tapia & Schöler, 2010*) as in the case of H103 which was able to maintain its pluripotent features under prolonged passage. Furthermore, it has been shown that TGF-β triggers epithelial-mesenchymal transition (EMT) (*Ebrahimi, 2015*) and the presence of transforming growth factor beta-I (TGF-β) signal transduction was previously reported to be among the root causes of roadblocks in reprogramming (*Li et al., 2010*). Moreover, successful reprogramming towards pluripotency was shown to be facilitated by mesenchymal-to-epithelial transition (MET) followed by suppression of epithelial-to-mesenchymal transition (EMT) regulation (*Chen, Han & Pei, 2012*). Paterson and colleagues determined the effect of transforming growth factor beta-I (TGF-β) in OSCC H-series cell lines in which H376 cell line was shown to be more *responsive to* TGF-β *than H103* (*Paterson et al., 1995*). As such, H376 may potentially harbour higher EMT activity which makes it more resistance towards reprogramming.

It was previously showed that extensive passaging enhances the reprogramming process (*Chin et al., 2009*) in which the hierarchical pluripotency gene activation takes place in a gradual manner upon prolong *in-vitro* passaging (*Shan et al., 2014*). Notably, in our study, down-regulation of Oct4 in P5 for Rep-H103 may indicate that its expression was not fully activated at the initial phase at P5 but its expression was more distinct at a later phase at P10 whereby more stable expression of pluripotent genes are expected to occur within the reprogramming process. Nevertheless, the fundamental pluripotency regulators (Oct4, Sox2 and Nanog) in Rep-H103 were expressed at higher levels than that of Rep-H376 at both passage 5 and passage 10. We observed that Nanog expression indicated a distinct difference between the two cell types and their susceptibility towards reprogramming. The interactions between Oct4, Sox2 and Nanog had been demonstrated via mutagenesis *in-vitro* assay and *in-vivo* functional study described previously in which suppression of Oct4 and Sox2 expression respectively decreases the promoter activity of Nanog (*Rodda et al., 2005*). Furthermore, the deficiency in Nanog expression results in partially reprogrammed cells which are unable to shift into pluripotency state due to impaired regulation of pluripotency network (*Festuccia et al., 2013*). Such was the case as reported by *Miyoshi et al. (2009)* in which gastrointestinal cell lines selected for reprogramming were shown to express low level of Nanog mRNA, but gradually acquired significant up-regulation of Nanog expression upon reprogramming with four pluripotent transcription factors. In our studies, further evidence of pluripotency was confirmed on successfully Rep-H103 via the immunofluorescence staining for common intracellular (Oct4, Sox2, Nanog) and extracellular (Tra-1-60) pluripotent markers, which were used in pluripotent stem cells characterization. All intracellular and extracellular pluripotent markers were detected on reprogrammed H103 indicating pluripotency expressions were achieved at the protein level.

As Klf4 is known to act either as a tumour suppressor gene or an oncogene, depending on the need of the tumour cells and the types of cancer (*Evans & Liu, 2008*) and c-Myc is a crucial oncogene that confers immortality in cancer cells via a shift from senescence state to oncogenic progression (*Erenpreisa & Cragg, 2013*), these transcription factors have been highly implicated in influencing cancer progression. The down-regulation pattern

of oncogenic gene expression, namely c-Myc and Klf4 in both H103 and H376 were also observed in reprogrammed human osteosarcoma cells (*Zhang et al., 2013*). Furthermore, when teratoma formations were assessed, it was found that the parental cancer cells formed tumours at a faster rate than that of the reprogrammed counterpart which exhibited a reduced aggressive cancer phenotype. Down-regulation of both c-Myc and Klf4 in reprogrammed OSCC suggests reprogramming may initiate an epigenetic reversal process on the oncogenic gene networks in cancer cells and this phenomenon could be utilized as a therapeutic strategy for treatment of OSCC.

The differentiation potential demonstrated that Rep-H103 which initially originated from ectoderm and endoderm lineage (*Jones & Klein, 2013*), portrays its capability of differentiating into osteocytes and adipocytes which are of the mesoderm lineage. The ability to differentiate into all three germ layers has been commonly observed in successfully reprogrammed cells including iPS-like cancer cells (*Carette et al., 2010*; *Rizzino, 2013*) and is crucial in generating post-iPSCs to unravel the underlying tumorigenesis development of the specific cancer.

Reprogrammed OSCC cell lines (H103 and H376) were used as a cancer-specific model to provide a conceptual study on the cells ability to be reprogrammed into a pluripotent state and their pluripotent characteristics at the in-vitro level. Since our study involves a stage I (H103) and stage III (H376) OSCC cell lines, the findings obtained do not address the effect of reprogramming on stage IV OSCC (the final stage in the STNMP classification) and the outcomes of the pluripotent signals at the in-vivo level. The next phase of the study is being planned to address the current limitation of the findings.

In summary, the maintenance of stem cell like morphology and pluripotent expressions upon reprogramming was only observed in Rep-H103 cells. This may be due to the differences in the inherent genetic make-ups between H103 and H376 cell lines, which determined their capacity to be reprogrammed into iPS-like cells. Our findings indicated a typical up-regulation pattern of endogenous transcription factors Oct4, Sox2 and Nanog in the reprogrammed H103 cell. However, down-regulation of oncogenes, c-Myc and Klf4 was observed in both Rep-H103 and Rep-376 cells. As such, the OSCC reprogrammed cells are potential models for further studies on cancer progression in OSCC by enabling access to cancer properties from the initial tumour initiation to the later malignant/metastatic states as well as models for the discovery of novel anti-cancer drugs.

## ACKNOWLEDGEMENTS

We would like take this opportunity to thank Assistant Professor Dr. Shigekii Sugii from DUKE-NUS Graduate Medical School, Singapore, for sharing his iPSCs knowledge and technology with us. We would also like to thank Dr. Teoh Hoon Koon, Teh Hui Xin and Choong Pei Fen who sequenced the plasmids.

### Funding

This work was supported by University of Malaya HIR-MOHE Grant Initiative (Reference number-UM.C/625/1/HIR/MOHE/CHAN/03) and University Tunku Abdul Rahman Research Funding (Vote number-UTARRF 6200/A11). The funders had no role in study design, data collection and analysis, decision to publish, or preparation of the manuscript.

### Grant Disclosures

The following grant information was disclosed by the authors:
University of Malaya HIR-MOHE Grant Initiative: Reference number-UM.C/625/1/HIR/MOHE/CHAN/03.
University Tunku Abdul Rahman Research Funding: Vote number-UTARRF 6200/A11.

### Competing Interests

The authors declare there are no competing interests.

### Author Contributions

- Nalini Devi Verusingam conceived and designed the experiments, performed the experiments, analyzed the data, wrote the paper, prepared figures and/or tables.
- Swee Keong Yeap conceived and designed the experiments, analyzed the data, reviewed drafts of the paper.
- Huynh Ky performed the experiments, contributed reagents/materials/analysis tools.
- Ian C. Paterson, Suan Phaik Khoo and Tunku Kamarul contributed reagents/materials/analysis tools.
- Soon Keng Cheong analyzed the data, reviewed drafts of the paper.
- Alan H.K. Ong conceived and designed the experiments, analyzed the data, contributed reagents/materials/analysis tools, reviewed drafts of the paper.

### Data Availability

The raw data has been supplied as a Supplementary File.

### Supplemental Information

Supplemental information for this article can be found online at http://dx.doi.org/10.7717/peerj.3174#supplemental-information.

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
