# Peer review of "Susceptibility of Human Oral Squamous Cell Carcinoma (OSCC) H103 and H376 cell lines to Retroviral OSKM mediated reprogramming"

_PeerJ, doi:10.7717/peerj.3174_

## Round 0.1 · original submission · Minor Revisions

I deeply apologize for big delay of the review process. We finally obtained reviewer's comments. All of them appreciate importance of your study, but some of the reviewer has criticism. I hope you find these comments valuable to improve your manuscript. I hope you submit the revise manuscript. If you need more than 40 days to resubmit revised manuscript, please let me know.

Reviewer 1 ·

Basic reporting

I suggest that authors need to define the abbreviations that they have used at the first time. For example, line 2 Abstract page; iPSC's.... OSCC...

Experimental design

Well-designed study.

Validity of the findings

No comments.

Additional comments

- This is a good study, and I believe the authors need to comment in the discussion section on the limitations of their study.

·

Basic reporting

No comments.

Experimental design

1. Under Material and Method section authors chose to select H103 (STNMP Stage I) and H376 (STNMP Stage III), Human Oral Squamous Cell Carcinoma (OSCC) cell lines. Why authors did not take stage IV (STNMP) OSCC cell line which is the last stage in the STNMP classification.

Validity of the findings

No comments.

·

Basic reporting

No Comments

Experimental design

iPSC could form teratoma in vivo. In the manuscript, the authors indicated that other research groups prefer agree reprogrammed cancer cells can form cancer stem cells. That means reprogrammed cancer cells induced malignant tumor but not teratoma in vivo. In this study, the authors did not demonstrate the results of in vivo study of the reprogrammed OSCC cells. So, the findings are not strongly convincing.

Validity of the findings

The in vitro findings are acceptable.

Additional comments

Please test the xenograft assay and resubmit the manuscript.

Reviewer 4 ·

Basic reporting

The manuscript by Verusingam attempts to examine the reprogramming ability of OSCC-derived cells (H103 and H376) by using OSKM mediated reprogramming method. They have used retroviral transduction method to express Oct4, Sox2, KLf4 and c-Myc genes in OSCC cells. Afterwards, morphological investigation, qRT-PCR and functional assays were performed to examine and characterize the generation of iPSCs. Although the paper is interesting, there are several issues which need to be clarified / revised.

Experimental design

Experimental design is optimal. Some of the comments related to the experimental design are listed in the 'General comments for the author' section.

Validity of the findings

No comments

Additional comments

Major comments:
1. To demonstrate the successful infection, qRT-PCR data for H103 and H376 cells for Oct4, Sox2, KLf4 and c-Myc genes need to be provided after 48-72 hrs after infection. This is especially important because the expression levels of Klf4 and c-Myc are found to be strangely reduced in the rep-cells.
2. Down-regulation of Oct 4 in P5 for rep-H103 needs more discussion.
3. Qualities of all of the figures are not good. It is very difficult to read the labels in the figures. Figures 1, 2, 4 and 5 have to be remade. Please use magnified images, especially for IF.
4. Discussion section is lengthy and needs to be reduced and streamlined.
5. Linguistic correction is needed.

Minor comments:
1. When describing the method section (retroviral vectors), presence of selection marker (GFP) should be mentioned.
2. How was the transduction efficiency calculated in the OSCC cells? By FACS or counting the GFP positive cells under fluorescent microscope?

---

## Round 0.2 · Minor Revisions

I found you response will be satisfactory to most reviewers comments. However, I must ask you to improve labeling of the figures as reviewer 4 suggested before acceptance.

Use larger font to label the figures 2B, 2C, 4, and 5 because they are still too small to read. In addition, scale bars are too small to see. Enlarge scale bars (use thicker lines) in figures 1, 2B, 2C. Add scale bars in figures 4A, 4B, 5A, 5B (all top panels if necessary) and 6.

---

## Round 0.3 · accepted · Accept

I am happy to see new figures with scale bars.